# Influence of Growth Medium Composition on Physiological Responses of *Escherichia coli* to the Action of Chloramphenicol and Ciprofloxacin

**DOI:** 10.3390/biotech12020043

**Published:** 2023-06-01

**Authors:** Galina Smirnova, Aleksey Tyulenev, Nadezda Muzyka, Vadim Ushakov, Zoya Samoilova, Oleg Oktyabrsky

**Affiliations:** Institute of Ecology and Genetics of Microorganisms, Perm Federal Research Center, Russian Academy of Sciences, Goleva 13, 614081 Perm, Russia; leksey333@yandex.ru (A.T.); mu2ykana@mail.ru (N.M.); ushakovvad@yandex.ru (V.U.); samzu@mail.ru (Z.S.); oktyabr@iegm.ru (O.O.)

**Keywords:** H_2_S, antibiotics, rich and minimal media, growth, respiration, bacterial survival

## Abstract

The ability of hydrogen sulfide (H_2_S) to protect bacteria from bactericidal antibiotics has previously been described. The main source of H_2_S is the desulfurization of cysteine, which is either synthesized by cells from sulfate or transported from the medium, depending on its composition. Applying electrochemical sensors and a complex of biochemical and microbiological methods, changes in growth, respiration, membrane potential, SOS response, H_2_S production and bacterial survival under the action of bactericidal ciprofloxacin and bacteriostatic chloramphenicol in commonly used media were studied. Chloramphenicol caused a sharp inhibition of metabolism in all studied media. The physiological response of bacteria to ciprofloxacin strongly depended on its dose. In rich LB medium, cells retained metabolic activity at higher concentrations of ciprofloxacin than in minimal M9 medium. This decreased number of surviving cells (CFU) by 2–3 orders of magnitude in LB compared to M9 medium, and shifted optimal bactericidal concentration (OBC) from 0.3 µg/mL in M9 to 3 µg/mL in LB. Both drugs induced transient production of H_2_S in M9 medium. In media containing cystine, H_2_S was produced independently of antibiotics. Thus, medium composition significantly modifies physiological response of *E. coli* to bactericidal antibiotic, which should be taken into account when interpreting data and developing drugs.

## 1. Introduction

A need to create new antibiotics and search for ways to enhance efficiency of existing antimicrobials in the face of a rapid increase in resistant pathogenic strains requires a comprehensive study of bacterial physiological responses under stress induced by antibiotics [1]. Unlike bacteriostatic drugs, which predominantly inhibit growth of bacteria, bactericidal antibiotics kill bacterial cells, thereby preventing a disease from becoming chronic and reducing the likelihood of the appearance of resistant strains. In 2007, the Collins group proposed a hypothesis according to which antibiotics with different intracellular targets kill bacteria by a single mechanism by increasing production of highly deleterious hydroxyl radicals during the Fenton reaction [2]. It is hypothesized that H_2_O_2_ involved in this reaction is formed in the respiratory chain upon acceleration of electron transport, and the concentration of free iron increases due to the destruction of FeS clusters during toxic changes in metabolism after the primary drug–target interactions. The hypothesis caused a wide resonance, since such a mechanism of bactericidal activity makes it possible to increase efficiency of existing antibiotics and design novel drugs capable of both influencing pathways of generation of reactive oxygen species (ROS) and altering levels of activity of antioxidant systems. However, the ROS-dependent mechanism of bactericidal activity has been challenged by several scientific groups, which have shown the absence of oxidative stress under the action of antibiotics [3,4,5]. Subsequent work by the Collins group provided further evidence for ROS involvement in bacterial killing, including analysis of respiratory activity and changes in H_2_O_2_, glutathione (GSH) and antioxidant gene expression [6,7,8]. They concluded that in the killing process ROS may be synergistic with the damage directly caused by the antibiotic in the primary target. In this case, factors that stimulate formation of ROS should increase sensitivity of bacteria to antibiotics. Endogenous and exogenous H_2_S has been reported to protect bacteria from bactericidal antibiotics and H_2_O_2_, supposedly through H_2_S-mediated free iron sequestration and prevention of the Fenton reaction [9,10,11]. Inactivation of H_2_S-producing enzymes, either by mutation or by inhibitors, made those bacteria highly susceptible to a variety of antibiotics [9,11].

Fluoroquinolone ciprofloxacin used in our work belongs to bactericidal antibiotics. It kills bacteria by damaging their DNA through direct binding with DNA gyrase and/or topoisomerase IV, which results in the formation of double-strand DNA breaks (DSBs), replication arrest, chromosome fragmentation and cell death [12]. ROS have been reported to be involved in the mechanism of quinolone-mediated rapid lethality [2,13]. However, in our own studies, we did not reveal an increase in sensitivity of *E. coli* to ciprofloxacin in mutants with defective redox systems of glutathione and thioredoxin, as well as in mutants lacking the general stress response regulator RpoS [14,15,16]. The effect of mutations and additives that change the redox situation in cells on bacterial survival was inversely related to their effect on the growth rate [16,17]. We did not observe ciprofloxacin-dependent stimulation of respiration, H_2_O_2_ formation, GSH oxidation and induction of antioxidant genes [15,16]. We also found that the effect of chloramphenicol and high concentrations of ciprofloxacin on *E. coli* was accompanied by a transient generation of H_2_S due to appearance of excess cysteine as a result of inhibition of protein synthesis [18,19,20]. An excess of cysteine is toxic to cells. Because of its high redox activity, cysteine is susceptible to autooxidation with ROS formation, and can also reduce Fe^3+^ to Fe^2+^, which potentiates the Fenton reaction in the presence of H_2_O_2_ [21,22]. Generation of H_2_S together with export of excess cysteine and its incorporation into glutathione are mechanisms of cysteine homeostasis that maintain its cytoplasmic concentration at a low level [19]. However, *E. coli* lacking cysteine synthetase B encoded by the *cysM* gene was equally sensitive to ciprofloxacin as the wild-type strain, despite the absence of H_2_S production in this mutant [16]. Discrepancies between our data and results of other researchers described above may be due to a strong dependence of ROS formation on cultivation conditions. All of our experiments were performed using M9 minimal glucose medium, while in most of the works of other authors rich LB medium was used. Differences in media composition can affect not only growth and metabolic rates, but also production of ROS. In contrast to minimal media, where cells need to synthesize cysteine using sulfate, an LB medium constituent cystine is directly imported into cells, which may affect the ability of *E. coli* to maintain cysteine homeostasis. Therefore, in this work, we aimed to compare changes in physiological parameters and H_2_S production in *E. coli* treated with ciprofloxacin or chloramphenicol in media with different cystine content.

## 2. Materials and Methods

### 2.1. Bacterial Strains and Growth Conditions

A strain of *Escherichia coli* BW25113 (wild-type) used in this study was from the Keio collection. A strain carrying the transcriptional gene fusion *sulA*(*sfiA*)::*lacZ* was created by transduction with P1 phage from strain DM4000 [23]. Three types of media were used to grow bacteria: (1) M9 minimal medium (Na_2_HPO_4_ 12H_2_O—15.13 g/L, KH_2_PO_4_—3 g/L, NH_4_Cl—1 g/L, NaCl—0.5 g/L, MgSO_4_ 7H_2_O—0.246 g/L, CaCl_2_—0.011 g/L) with glucose (0.2%) [24]; (2) M9 medium without sulfate, but with glucose (0.2%), casamino acids (CA, 0.2%) and feeding 15 µM cystine every 45 min; (3) Luria–Bertani (LB Miller) medium (pepton—10 g/L, yeast extract—5 g/L, NaCl—10 g/L). Cultures were grown overnight, then centrifuged, diluted in 80 mL fresh medium to an initial optical density at 600 nm (OD_600_) of about 0.05 and grown in 250 mL flasks with shaking (150 rpm) at 37 °C. When OD_600_ reached 0.4 (mid-exponential phase), chloramphenicol (Cam, 25 μg/mL) or ciprofloxacin (CF) at concentrations of 0.03 (2 × MIC), 0.3, 3 and 10 μg/mL was added to the medium and incubation continued for two hours. The specific growth rate (µ) was calculated by equation µ = Δln OD_600_/Δt, where *t* is the time in hours.

### 2.2. Real-Time Monitoring of Dissolved Oxygen (dO_2_), pH and Extracellular K^+^ and Sulfide

Dissolved oxygen and pH in *E. coli* cultures were continuously measured directly in the flasks using a Clarke oxygen electrode InPro 6800 (Mettler Toledo, Greifensee, Switzerland) and a pH electrode ESC-10601/7 (“IT” Company, Moscow, Russia), respectively. The dO_2_/pH controller of a BioFlo 110 fermentor (New Brunswick Scientific Co., Edison, NJ, USA) was used for data recording.

Extracellular sulfide was continuously recorded directly in the flasks using a system of sulfide-specific ion-selective chalcogenide XC-S^2^-001 (operating pH range 6–12) (Sensor Systems Company, St. Petersburg, Russia) and reference electrodes and a computer pH/ion meter cpX-2 (IBP, Pushchino, Russia).

Changes in levels of extracellular K^+^ were registered using a system of K^+^-selective (ELIS-121K) and reference electrodes. For a sensitive determination of K^+^ during *E. coli* growth in M9 and M9 + CA + cystine media, potassium concentration was reduced to 0.2 mM. Synchronous processing of all primary data from the sensor system was carried out using the RS-232 and Modbus protocols and the Advantech OPC Server v3.0 software package.

### 2.3. Determination of ATP, NAD^+^/NADH Ratio and Membrane Potential

ATP concentration was measured using a luciferin–luciferase ATP determination kit (Molecular Probes). Samples for analysis were removed before antibiotic addition and then every 30 min for 2 h. Then, 50 µL of the cell suspension was added to 450 µL of cell disruption reagent dimethyl sulfoxide (DMSO) to extract ATP. After 5 min, ATP concentration was determined according to the manufacturer’s protocol.

NAD^+^ and NADH pools and NAD^+^/NADH ratio were determined by the recycling assay as described previously [25]. Samples were removed immediately before drug addition (time zero) and after 25 min of exposure. Sample preparation and analysis were performed as described previously [16].

Membrane potential changes (∆ψ) were assessed using the ∆ψ-sensitive fluorescent dye DiBAC_4_(3), which stains only depolarized cells [26], as described elsewhere [16]. Samples of log-phase cells treated with protonophore carbonylcyanide *m*-chlorophenylhydrasone (CCCP, 20 µM) were used as positive control. Fluorescent cells were counted using a Leica DM2000 microscope as earlier described [16]. Total cell numbers were counted in transmitted light. All experiments were conducted 3–6 times on separate days.

### 2.4. Determination of H_2_S in the Gas Phase

Formation of gaseous H_2_S was assessed using paper strips soaked with lead acetate [Pb(Ac)_2_], which specifically reacts with H_2_S to form a brown stain of lead sulfide. The paper strips were fixed above the surface of the liquid culture. Spots were scanned and quantified using ImageJ software. The mean values of histograms of color intensity of each paper strip were measured. Data were presented as the ratio of the value before antibiotic treatment to the values obtained after 2 h of exposure, expressed as a percentage. All experiments were performed 3–6 times on separate days.

### 2.5. Study of Cell Viability and β-Galactosidase Activity

For colony-forming studies, culture samples taken before and 0.5, 1 and 2 h after antibiotic addition were washed (centrifuged for 2 min at 15,000 *g* and resuspended in an equal volume of 0.9% NaCl), serially diluted in 0.9% NaCl solution, mixed with molten soft LB-agar (0.8%) at 42 °C and poured onto plates with solid LB-agar (1.5%). Colonies were counted after 24 h of incubation at 37 °C.

The genotoxic effect of ciprofloxacin was assessed by the degree of expression of the *sulA* gene, which belongs to the SOS regulon and is induced in response to DNA damage [27]. Changes in *sulA* expression were determined in *E. coli* strain carrying the *sulA*::*lacZ* gene fusion by measuring β-galactosidase activity by Miller’s method [24].

### 2.6. Statistical Analysis of the Data

Each result is indicated as the mean value of three to six independent experiments ± the standard error of the mean (SEM). Significant difference was analyzed by Student’s *t*-test. A p value of 0.05 was used as the cut-off for statistical significance. Results were analyzed by means of the program packet Statistica 8.0.360 (StatSoft Inc., Tulsa, OK, USA, accessed on 27 August 2007). The data obtained using the electrodes represent the results of one of at least three independent experiments.

## 3. Results

### 3.1. Effect of Chloramphenicol and Ciprofloxacin on the Growth and Respiration of E. coli during Cultivation in Different Media

In our experiments, bacteria were cultivated in media widely used both for experimental studies and for biotechnological purposes. These media differ significantly in composition, including the source of sulfur. M9 glucose minimal medium contains sulfate as the only sulfur source. When sulfate was removed, but M9 was supplemented with casamino acids and cystine (M9 + CA + cys), cystine became the main source of sulfur. The standard rich LB medium (2.5 g per 100 mL H_2_O) contains cystine (156 ± 3 μM cysteine) and glutathione (11.2 ± 0.6 μM GSH) according to our analyses. When grown in these media, bacteria can differ both in growth rate and in ability to produce H_2_S, the source of which is the desulfurization of cysteine, which is either synthesized by cells or imported from the medium. We studied how composition of the growth medium affects action of antibiotics chloramphenicol and ciprofloxacin on *E. coli* growth and respiration.

Bacteriostatic action of chloramphenicol is due to inhibition of protein synthesis via binding to the 50S ribosomal subunit. Fluoroquinolone ciprofloxacin is a bactericidal antibiotic that kills bacteria by damaging their DNA. In the absence of antibiotics, specific growth rate values of *E. coli* BW25113 (µ_max_) were 0.68 ± 0.015, 1.18 ± 0.02 and 1.57 ± 0.05 h^–1^ when cultivated in M9, M9 + CA + cys and LB media, respectively. Regardless of type of the medium used, addition of chloramphenicol resulted in a sharp growth inhibition during the first 15 min (Figure 1). The effect of ciprofloxacin on *E. coli* growth rate depended on antibiotic concentration and medium type. A significant decrease in μ occurred at 0.03 μg/mL CF in M9, at 0.3 μg/mL CF in M9 + CA + cys and at 3 μg/mL CF in LB (Figure 1). A dose of 3 μg/mL CF caused a sharp decrease in the growth rate in all media except LB, and after the 75th minute μ took negative values, indicating a decrease in biomass density. In LB medium, this effect was achieved after 105 min of exposure to ciprofloxacin. At 10 μg/mL CF, a sharp decrease in μ occurred in all media, and its negative values were reached after 60 min of antibiotic exposure. In general, the richer the medium and the higher the initial growth rate of bacteria were, the longer the high levels of μ and, consequently, the high metabolic activity of cells retained during exposure to ciprofloxacin.

Analysis of changes in respiration revealed a similar dependence on the type of medium and antibiotic dose. In the absence of antibiotics, the level of dissolved oxygen (dO_2_) gradually decreased with increasing biomass, despite constant rotation of the flasks (Figure 2a). The rate of this decrease, expressed as dO_2_/OD_600_∙min, was 1.49 ± 0.2, 2.7 ± 0.2 and 2.88 ± 0.13 upon growth in M9, M9 + CA + cys and LB medium, respectively.

A striking feature of LB medium was a sharp reversible inhibition of respiration and a corresponding surge in oxygen content when dO_2_ reached 30–40%. We assume that the observed phenomenon is a reflection of the transient process associated with the switch of respiratory metabolism to another substrate in a complex multicomponent LB medium, where catabolized amino acids are sequentially consumed [28].

Addition of chloramphenicol dramatically inhibited respiration in all media (Figure 2b). According to our previous studies [15,16,20], ciprofloxacin caused phase changes in *E. coli* respiration. In the first phase, oxygen consumption mode depended on the type of medium and antibiotic concentration and correlated with the ciprofloxacin effect on growth rate values. A dose of 0.03 μg/mL gradually inhibited oxygen consumption in M9 but had no effect in LB medium (not shown). After treatment with 0.3 μg/mL CF in M9, *E. coli* cells continued to consume oxygen for 30 min, after which the drop in dO_2_ stopped, while in LB medium this dose of ciprofloxacin had no effect, and in M9 + CA + cys medium slowed down oxygen consumption compared to the control (Figure 2c). Treatment with 3 μg/mL CF in LB medium affected respiration similarly to 0.3 μg/mL in M9 (Figure 2d). A dose of 3 µg/mL CF in M9 and M9 + CA + cys media caused a sharp increase in dO_2_ due to inhibition of respiration; a similar increase in dO_2_ in LB was observed only at 10 µg/mL CF (Figure 2d).

At ciprofloxacin concentrations of 0.3 and 3 µg/mL in M9 and M9 + CA + cys media and 3 and 10 µg/mL in LB medium, the second phase of inhibition of respiration was observed after the first phase (Figure 2c,d). The time of onset of complete inhibition of respiration coincided with transition of specific growth rate to negative values (Figure 1). Thus, the profiles of changes in dO_2_ under action of ciprofloxacin in different media showed that cells in a rich medium retain metabolic activity at higher concentrations of ciprofloxacin than in a poor minimal medium, which is in good agreement with the maintenance of a higher growth rate under these conditions.

### 3.2. Medium Composition Affects Ciprofloxacin-Induced Changes in E. coli Energetics

Glucose consumption during aerobic growth of *E. coli* in minimal media is accompanied by accumulation of acidic by-products, which leads to a decrease in pH. Accordingly, sensitive pH recording allows real-time tracking of changes in glucose consumption. In the absence of antibiotics, a gradual decrease in pH was observed in all media studied (Figure 3). In LB medium, pH decreased for 3 h after inoculation with bacteria, and then its gradual increase was observed, which, apparently, was caused by a transition to the use of amino acids as an energy source. Chloramphenicol significantly slowed down acidification in M9 and M9 + CA + cys media and completely stopped it in LB (Figure 3). CF suppressed glucose uptake and pH decrease in a dose-dependent manner. As with dO_2_ and growth rate, higher concentrations of ciprofloxacin were required to inhibit acidification in LB medium than in M9 and M9 + CA + cys media (Figure 3). 

In particular, a dose of 0.3 μg/mL CF reduced the rate of acidification in M9 and M9 + CA + cys media, but did not cause changes relative to the control in LB.

Changes in membrane potential under the action of antibiotics were assessed in two ways: by continuous recording of K^+^ in the medium using a K^+^ selective electrode and by fluorescent microscopy after bacteria were treated with the fluorescent dye DiBAC_4_(3), which carries a negative charge and stains only depolarized cells. The number of depolarized cells in control cultures changed little during 2 h of growth and was 1 ± 0.15, 2.6 ± 0.2 and 6.2 ± 0.4% in M9, M9 + CA + cys and LB media, respectively (Figure 4a and Appendix A). Chloramphenicol slightly affected membrane potential in M9 medium, increasing the number of stained cells to 3.8%, while in M9 + CA + cys and LB media, the proportion of depolarized cells increased to 24 and 30%, respectively (Figure 4a). Moreover, 0.3 and 3 µg/mL ciprofloxacin increased the number of depolarized cells in all media studied, with a maximum effect in LB (Figure 4b and Appendix A).

When growing bacteria in all media, a decrease in the extracellular concentration of K^+^ was observed due to its uptake by cells (Appendix A). Treatment with chloramphenicol was accompanied by a short-term acceleration of K^+^ uptake by cells, followed by a slowdown in the decline in its concentration compared to the control (Appendix A). Furthermore, 0.3, 3 and 10 μg/mL CF dose-dependently stimulated the release of K^+^ from cells after 60 min of exposure to the antibiotic in all studied media, which indicates the loss of membrane potential by cells (Appendix A–d). In general, both DiBAC fluorescence and extracellular K^+^ monitoring showed that exposure to ciprofloxacin in all media was accompanied by a drop in membrane potential.

Important indicators characterizing the energy states of cells are the ratio of NAD^+^/NADH and the level of ATP. According to the hypothesis of Kohanski et al. [2], bactericidal antibiotics accelerate respiration, resulting in a decrease in the NADH pool and, accordingly, an increase in the NAD^+^/NADH ratio. We have previously shown that 3 μg/mL CF in M9 medium with glucose causes a two-fold increase in the NAD^+^/NADH ratio [16]. Under the action of 10 μg/mL CF in LB medium, we also observed a slight decrease in the NADH pool and statistically significant increase in the NAD^+^ pool and the NAD^+^/NADH ratio (Appendix A). However, bacteriostatic chloramphenicol had the same effect, with both antibiotics dramatically inhibiting respiration at the doses used.

ATP levels in exponentially growing cultures of *E. coli* BW25113 were 2.65 ± 0.13, 2.17 ± 0.09 and 1.7 ± 0.09 µM/OD_600_ in M9, M9 + CA + cys and LB medium, respectively, and changed little during growth in the absence of antibiotics (Figure 5a and Appendix A). The action of both chloramphenicol and ciprofloxacin was accompanied by an increase in the intracellular concentration of ATP in all media (Figure 5). In contrast to Cam and 0.3 μg/mL CF, high concentrations of CF (3 and 10 μg/mL) after an initial increase in the level of ATP led to its decrease. Duration of exposure until the beginning of the fall in ATP levels decreased with increasing dose of CF (Figure 5b–d and Appendix A). The decrease in ATP coincided with the inhibition of respiration and the loss of membrane potential.

### 3.3. Sulfide Production under the Action of Antibiotics in Different Media

Previously, we found that chloramphenicol and ciprofloxacin induced H_2_S production in *E. coli* growing in M9 medium due to the appearance of a transient excess of intracellular cysteine [18,19,20]. The aim of this work was to determine how presence of cystine in the medium affects H_2_S formation under *E. coli* exposure to these antibiotics.

Changes in H_2_S level in the liquid medium were monitored using a sulfide electrode, and also in the gas phase by using lead acetate soaked paper strips. The sulfide sensor detects changes in H_2_S in the nanomolar range (from 10 nM), while the color of the paper strips appears when 0.1 μM Na_2_S is added to the cell-free medium. During control growth in M9 medium, *E. coli* cells did not release sulfide, and when Cam or 3 and 10 µg/mL CF were added, they induced H_2_S production, which was recorded as a reversible drop in the electrode potential and a change in the color of the paper strips (Figure 6a,d). The sulfide sensor showed higher sensitivity to changes in sulfide concentration than the strips. In particular, an increase in the level of H_2_S in the gas phase was observed only at 10 µg/mL CF and was absent at 3 µg/mL. When *E. coli* was grown in M9 medium without sulfate, but with CA and cystine, active production of sulfide began immediately after cell inoculation into the medium (Figure 6b). The drop in the potential of the sulfide electrode was irreversible and insensitive to subsequent treatment of cells with chloramphenicol and ciprofloxacin. In this medium, lead acetate soaked paper strips stained to the limit of sensitivity (about 500%) and did not change color when *E. coli* was exposed to antibiotics. Control cultures of *E. coli* growing in LB medium also produced sulfide without any external influences (Figure 6c). A sharp drop in the sulfide electrode potential coincided in time with a sharp reversible increase in dO_2_ (Figure 2a), which gave evidence of a transient process associated with metabolic rearrangement. At the same time, accumulation of H_2_S in the gas phase was recorded (Figure 6d). Addition of antibiotics in the zone of the maximum drop in the electrode potential, which corresponded to the maximum production of sulfide, had no effect (not shown). Treatment of *E. coli* with Cam or CF in the phase preceding onset of its own production of H_2_S (OD_600_ about 0.3) modified the mode of change in the electrode potential, affecting its amplitude and duration depending on the antibiotic concentration (Figure 6c). The amount of H_2_S determined in the gas phase also changed during 2 h of incubation with antibiotics (Figure 6d). Thus, in media containing cystine, an excess accumulation of intracellular cysteine can occur without external influences. This may affect the action of antibiotics, since cysteine can increase production of ROS, creating an additional burden on antioxidant systems.

### 3.4. Influence of Medium Composition on SOS Response and Viability of E. coli upon Exposure to Ciprofloxacin

DNA damage induced by quinolones activates the SOS gene network controlled by RecA/LexA, resulting in production of various repair proteins and increased tolerance to ciprofloxacin [27,29]. To assess a degree of SOS regulon expression under action of ciprofloxacin in different media, the *lacZ* gene fusion with the promoter of the SOS-controlled *sulA* gene was used. Activity levels of β-galactosidase at OD_600_ 0.4 were 220 ± 20, 795 ± 65 and 93 ± 4 Miller units when *E. coli* was grown in M9, M9 + CA + cys and LB media, respectively. In the absence of ciprofloxacin, the level of β-galactosidase did not change significantly during growth in M9 medium and gradually increased in M9 + CA + cys and LB media. Expression of *sulA*::*lacZ* in M9 medium 60 min after the drug treatment increased by factors of 6, 11 and 3 upon exposure to 0.03, 0.3 and 3 μg/mL CF, respectively (Figure 7). At a high initial level of *sulA*::*lacZ* expression in M9 + CA + cys medium, treatment with ciprofloxacin caused only a slight additional induction at 0.03 and 0.3 μg/mL CF. Level of *sulA*::*lacZ* expression in *E. coli* growing in LB medium increased by factors of 22, 30 and 36 at 0.03, 0.3 and 3 µg/mL CF, respectively (Figure 7). Thus, the maximum induction of the *sulA* gene was achieved at 0.3 µg/mL CF in M9 and M9 + CA + cys media and at 3 µg/mL CF in LB medium. At the same antibiotic concentrations, the maximum number of filaments was recorded in microscopic studies, which confirms the results obtained with the *sulA*::*lacZ* fusion. The observed high level of *sulA*::*lacZ* expression in control cultures growing in M9 + CA + cys medium may indicate an increase in DNA damage under these conditions.

The ability of antibiotics to kill bacteria in different media was studied by determining number of colony-forming units (CFU). As expected, chloramphenicol stopped the increase in CFU but did not kill the cells, regardless of the medium used (Appendix A). The effect of ciprofloxacin on survival was dependent on antibiotic concentration and medium type. At 0.03 μg/mL CF, the largest decrease in CFU (by two orders of magnitude) was observed in M9 medium, was less pronounced in M9 + CA + cys medium and was completely absent in LB (Figure 8a). However, with an increase in the dose of ciprofloxacin (0.3, 3 and 10 μg/mL), bacteria growing in M9 + CA + cys and LB media showed greater sensitivity than in M9 (Figure 8b,c and Appendix A). During the fast phase of cell death under the action of 3 μg/mL CF, the difference in CFU between M9 and LB media reached three orders of magnitude.

It is known that bactericidal activity of quinolones, including ciprofloxacin, is a two-phase process: the lethality of drugs increases to a certain optimal concentration, after which it decreases [30]. The ability of bacteria to maintain higher metabolic activity in rich media was accompanied by a shift in the optimal bactericidal concentration (OBC) of ciprofloxacin towards higher values. The graph of logCFU versus ciprofloxacin concentration shows that in M9 medium OBC is 0.3 µg/mL, in LB medium it is 3 µg/mL, and in M9 + CA + cys medium it is in the range between these two values (Figure 8d). Based on the *sulA* gene expression data, maximum SOS regulon induction was observed at ciprofloxacin concentrations corresponding to the OBC for each of the media (Figure 7).

## 4. Discussion

Taken together, the data obtained in this work showed that the main difference in physiological response of *E. coli* to bacteriostatic and bactericidal antibiotics is the rate of inhibition of metabolic processes, regardless of the culture medium used. In all the media studied, the protein synthesis inhibitor chloramphenicol caused a rapid inhibition of growth and respiration, slowed K^+^ uptake and acidification of the medium associated with glucose consumption and also increased the NAD^+^/NADH ratio and ATP level. In contrast to M9, in M9 + CA + cys and LB media, chloramphenicol significantly reduced membrane potential measured with DiBAC_4_(3) fluorescent dye.

Fluoroquinolone ciprofloxacin binding to both gyrase and DNA and stabilizing the gyrase-DNA-cleaved complex was shown to inhibit DNA supercoiling and relaxation [12,29]. Apparently being not rapidly lethal at this stage, it inhibited replication and transcription and induced SOS response. However, it was found that, after gyrase removing from DNA, double-strand DNA breaks were formed, which could lead to chromosome fragmentation and cell death. In our study, ciprofloxacin inhibited *E. coli* growth, respiration, acidification and K^+^ uptake, and increased ATP levels and the NAD^+^/NADH ratio in a dose-dependent manner. Degree of this inhibition depended on the medium used: in the richer M9 + CA + cys and LB media, higher concentrations of ciprofloxacin were needed to completely inhibit metabolism than in the minimal M9 medium. The inhibition of respiration caused by ciprofloxacin was biphasic in all media. At ciprofloxacin concentrations not exceeding OBC, a phase was observed in which the cells continued totake up oxygen. The duration of this phase coincided with the time during which the cells maintained a high growth rate. Concentrations of ciprofloxacin above OBC caused a rapid inhibition of respiration already in the first phase. The onset of the second phase of inhibition of respiration depended on the dose of ciprofloxacin and earlier was associated with the onset of the SOS response late phase (“programmed cell death”) and was absent in *recA* mutant [15,16]. The beginning of this phase coincided with the complete cessation of growth and consumption of the energy source, the release of K^+^ from the cells, and a drop in the membrane potential and the ATP pool. We did not observe an acceleration of oxygen consumption in any of the studied media under the action of all concentrations of ciprofloxacin. This is consistent with our previous data for ciprofloxacin and the results obtained by Liu and Imlay for the fluoroquinolone norfloxacin [4,15,16,20], but is not consistent with the data of the Collins group, which reported an increase in respiration rate under the action of norfloxacin [7]. However, under the action of bactericidal ciprofloxacin, high metabolic activity persisted longer than under the action of a bacteriostatic chloramphenicol. The greatest similarity in the physiological responses of bacteria to ciprofloxacin and chloramphenicol was observed at doses of ciprofloxacin above the OBC.

At the same concentration of ciprofloxacin, *E. coli* maintained a higher rate of growth and respiration in LB than in M9 medium. The specific growth rate is an integral parameter, which reflects the activity of core metabolic processes and, accordingly, the number of active targets for antibiotics [31]. Therefore, there are more ciprofloxacin targets in cells and more DNA damage (stalled transcriptional complexes and replication forks and DSBs) accumulates in LB than in M9 medium at a given dose of ciprofloxacin. The SOS system can be involved in reparation of damage induced by quinolones, and as *E. coli* can successfully repair only up to four simultaneous DSBs, slowly growing cells with fewer replication forks and reduced transcriptional activity experience fewer breaks and are able to repair them and survive [29]. Fast growing cells receive more damage and die, which explains the inverse relationship between CFU and specific growth rate and the decrease in the number of CFU by 2–3 orders of magnitude in LB compared to M9 medium. This is consistent with our earlier data that, regardless of the causes affecting the growth rate, even a small increase in μ is accompanied by a significant increase in ciprofloxacin-induced cell death [17]. In addition, the maintenance of respiratory activity during drug-induced stress may promote the formation of ROS, which can potentially damage cellular structures and contribute to the lethal activity of ciprofloxacin. That is, a paradox arises: the less sensitive the metabolic processes of cells in liquid culture are to the action of ciprofloxacin, the stronger the lethality in the CFU test. The fast phase of cell death on the killing curves obtained by counting CFU corresponds to a period of high metabolic activity of cells in liquid culture. From this point of view, the late phase of the SOS response, which completely stops metabolism and transfers cells to a dormant state, may be an adaptive response that increases the number of persisters. Our data are consistent with Hong et al.’s findings that most quinolone-induced cell death seems to occur not during antibiotic exposure, but at the growth recovery stage on LB agar plates in the absence of antibiotics [13], although the dominant role of ROS in this process needs further confirmation.

Another consequence of the slower inhibition of metabolism in rich media was a shift in the optimal bactericidal concentration towards higher doses of ciprofloxacin (from 0.3 µg/mL in M9 to 3 µg/mL in LB). The biphasic dependence of bactericidal activity on drug concentration, which is characteristic of quinolones [30], disappeared in the *recA* mutant [15]. In this work, we have shown that the maximum expression of the *sulA* gene, i.e., the maximum induction of the SOS response, corresponds to the OBC for ciprofloxacin in different media. It is possible that the dramatic inhibition of growth and respiration and the resulting reduction in DNA damage and increased survival at quinolone concentrations above OBC are controlled by the SOS system. Previously, it was suggested that the increased bactericidal activity of moderate concentrations of quinolones compared to their high doses may be associated with a decrease in ROS production at high doses of antibiotics [32]. When respiration is inhibited, ROS production may decrease. However, at the same time, under the action of high doses of quinolones, there is a decrease in replication and transcription, and, accordingly, the number of active targets for antibiotics decreases, which can block antibiotic action.

Here and in our previous work [18,19,20], we have shown that treatment of exponentially growing *E. coli* with chloramphenicol or high doses of ciprofloxacin (3–20 µM/mL) in M9 medium is accompanied by transient H_2_S generation. Hydrogen sulfide formation has also been observed during starvation stress caused by glucose depletion or valine supplementation (isoleucine starvation) [18,19]. We found that the production of H_2_S is a marker of the appearance of excess free cysteine in the cytoplasm during a sharp inhibition of protein synthesis [18,19]. Sulfide generation, as well as the acceleration of glutathione synthesis and the export of excess cysteine from cells, contributed to the reduction of cysteine level in the cytoplasm and can be considered as mechanisms of its homeostasis. Maintaining a low intracellular level of cysteine is very important for cells because cysteine is capable of autoxidation to form ROS, and can also reduce Fe^3+^ to Fe^2+^, potentiating the Fenton reaction to form deleterious hydroxyl radicals [21,22]. It can be assumed that under the action of bactericidal antibiotics, ROS formed when excess intracellular cysteine appears will contribute to the lethality of the antibiotic. However, *E. coli* transiently generated H_2_S in M9 medium only when exposed to ciprofloxacin concentrations above OBC, when growth inhibition was severe. Under these conditions, inhibition of H_2_S production in the *cysM* mutant encoding cysteine synthase B did not affect survival compared to the parental strain [16]. In contrast to the M9 medium, when *E. coli* was grown in LB or M9 + CA + cys media, H_2_S was generated in the absence of external influences. Under these conditions, cysteine synthesis is almost completely inhibited [33], and *E. coli* cells are supplied with cysteine through cystine transport via CysB-controlled TcyP and TcyJLN importers [22,34]. Massive uptake of cystine by cells can create an excess of intracellular cysteine and trigger the mechanisms of its homeostasis, in particular H_2_S production and cysteine efflux. Thus, the effect of antibiotics on *E. coli* in LB and M9 + CA + cys media occurs in the background of increased intracellular cysteine and the already running mechanisms of cysteine homeostasis, which may significantly modify the stress response to antibiotics. We observed an increased expression of the *sulA* gene in the M9 + CA + cys medium before treatment of bacteria with ciprofloxacin, which could be a consequence of ROS-mediated DNA damage with an excess of intracellular cysteine. An alternative reason for increased *sulA* expression may be a change in the activity of the SOS response regulator RecA, which contains essential SH groups [35], during cystine-induced disulfide stress. The produced H_2_S is an anti-inducer of CysB and can inhibit the cysteine regulon [33]. In addition, low micromolar levels of sulfide have been shown to inhibit cytochrome *bo* oxidase and switch respiration to *bd* oxidase [36,37]. It was also found that H_2_S reduced susceptibility of bacteria to bactericidal antibiotics and H_2_O_2_ in the LB medium, presumably by reducing the level of intracellular Fe^2+^ and preventing the Fenton reaction [9,10]. Excess cysteine exported from cells can be oxidized to cystine in the presence of oxygen and then again uptake by the cells. Therefore, in the presence of cystine in the medium, a futile import/reduction/export cycle occurs that consumes a large amount of cellular energy [22,34], which may also affect antibiotic tolerance. In light of the above, the mechanisms of cysteine homeostasis are a convenient target for the development of new drugs that increase the efficiency of bactericidal antibiotics. Recently, inhibitors of bacterial H_2_S biogenesis have been created that reduce resistance and tolerance to antibiotics [11]. Our data on H_2_S production in different media indicate that such inhibitors will be more effective in rich media containing cystine/cysteine and less effective in minimal media where H_2_S production is short-lived.

## 5. Conclusions

Medium composition has little effect on the chloramphenicol-induced stress response in *E. coli*. Under action of ciprofloxacin in rich media, cells retain a high rate of growth and respiration longer than in a minimal medium, which results in a shift in the optimal bactericidal concentration from 0.3 μg/mL in M9 to 3 μg/mL in LB medium and a decrease in survival by 2–3 orders of magnitude in LB in comparison with M9 under treatment with the same dose of ciprofloxacin. Despite the general similarities in stress responses to ciprofloxacin of *E. coli* grown in minimal M9 and rich LB media, there are fundamental differences associated with media composition and peculiarities of intracellular cysteine homeostasis in the presence of high concentrations of exogenous cystine. This might be the reason for significant inconsistencies between the results obtained in different media. Thus, medium composition affects both bacterial growth rates and H_2_S production, which can significantly modify the tolerance of bacteria to antibiotics and must be taken into account when interpreting research data, as well as when developing and using drugs in medicine and biotechnology.

## Figures and Tables

**Figure 1 biotech-12-00043-f001:**
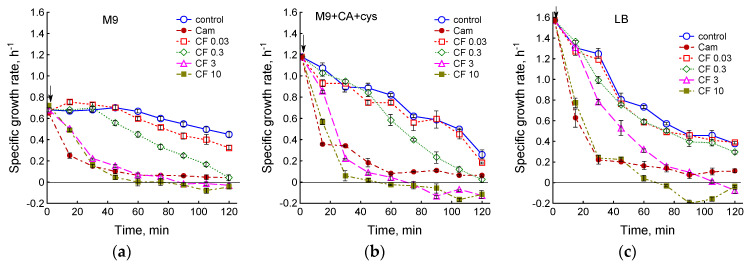
Effect of medium composition on the specific growth rate of *E. coli* under the action of chloramphenicol (Cam, 25 μg/mL) and ciprofloxacin (CF, 0.03, 0.3, 3 and 10 μg/mL). (**a**) Minimal medium M9 with sulfate; (**b**) M9 medium without sulfate, supplied with CA and cystine; (**c**) LB medium. Antibiotics were added at an OD_600_ of 0.4 at the time indicated by the arrow. Values are means and standard error (vertical bars) from at least three independent experiments.

**Figure 2 biotech-12-00043-f002:**
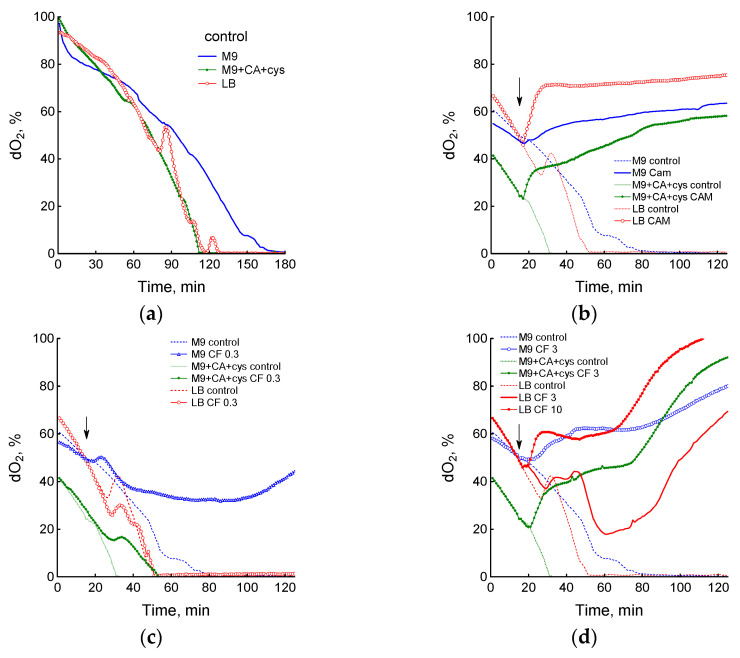
The mode of antibiotic-mediated changes in *E. coli* respiration depends on the composition of the medium and the drug concentration. (**a**) Dissolved oxygen change in the absence of antibiotics; (**b**) 25 µg/mL chloramphenicol (Cam); (**c**) 0.3 µg/mL ciprofloxacin (CF); (**d**) 3 μg/mL CF in all media and 10 μg/mL CF in LB. Antibiotics were added at an OD_600_ of 0.4 at the time indicated by the arrow. The data shown are representative.

**Figure 3 biotech-12-00043-f003:**
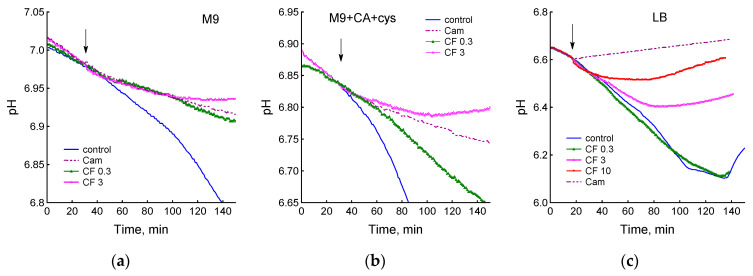
Changes in pH after *E. coli* treatment with chloramphenicol (25 µg/mL) or ciprofloxacin (0.3, 3 and 10 µg/mL). (**a**) M9 medium; (**b**) M9 + CA + cys medium; (**c**) LB medium. Antibiotics were added at an OD_600_ of 0.4 at the time indicated by the arrow. The data shown are representative.

**Figure 4 biotech-12-00043-f004:**
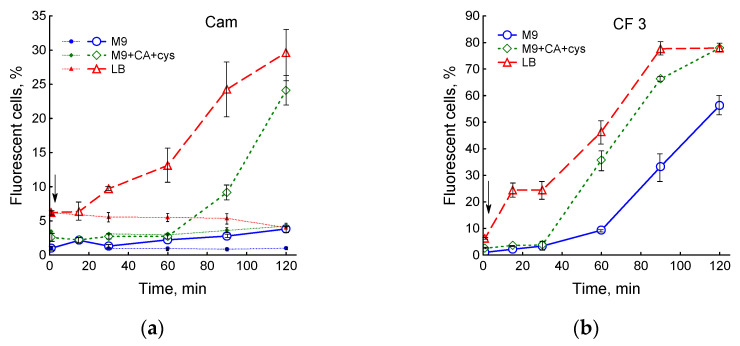
Medium composition affects the changes in the membrane potential of *E. coli* under the action of 25 µg/mL chloramphenicol (**a**) and 3 µg/mL ciprofloxacin (**b**). (**a**) Small symbols and dotted lines—control. Antibiotics were added at an OD_600_ of 0.4 at the time indicated by the arrow. Values are means and standard error (vertical bars) from at least three independent experiments.

**Figure 5 biotech-12-00043-f005:**
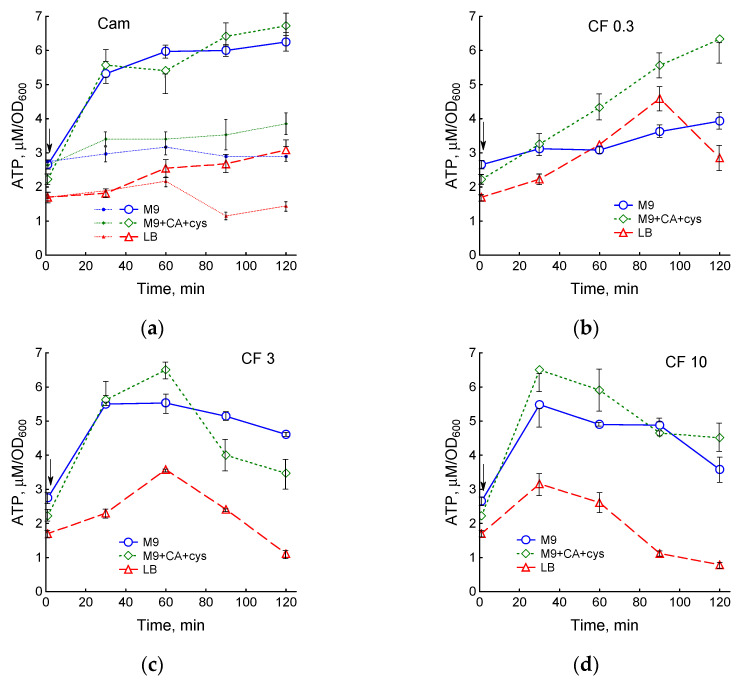
Changes in ATP levels during treatment of *E. coli* with chloramphenicol or ciprofloxacin in various media. (**a**) small symbols and dotted lines—control. Then, 25 μg/mL chloramphenicol (**a**), 0.3 μg/mL CF (**b**), 3 μg/mL CF (**c**) and 10 μg/mL CF (**d**) were added at an OD_600_ of 0.4 at the time indicated by the arrow. Values are means and standard error (vertical bars) from at least three independent experiments.

**Figure 6 biotech-12-00043-f006:**
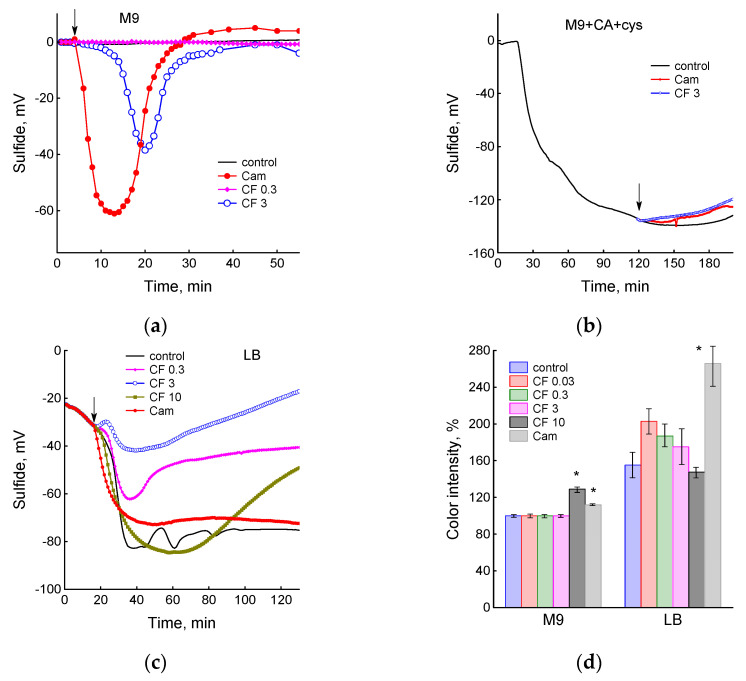
Sulfide production by *E. coli* cells in media of different composition. (**a**) Treatment of *E. coli* with chloramphenicol and high doses of ciprofloxacin in M9 medium stimulates reversible sulfide production. (**b**) In M9 + CA + cystine medium, continuous production of sulfide by cells is observed and antibiotic treatment does not affect this process. (**c**) In LB medium, *E. coli* BW25113 start to spontaneously produce sulfide at OD_600_ above 0.3, antibiotic treatment modifies this process. (**d**) Accumulation of H_2_S in the gas phase during two hours of exposure to antibiotics. (**a**–**c**) Antibiotics were added at an OD_600_ of 0.4 at the time indicated by the arrow. The data shown are representative. (**d**) Values are means and standard error (vertical bars) from at least three independent experiments. Statistical differences compared to control for each medium (*p* < 0.01) are noted with asterisks.

**Figure 7 biotech-12-00043-f007:**
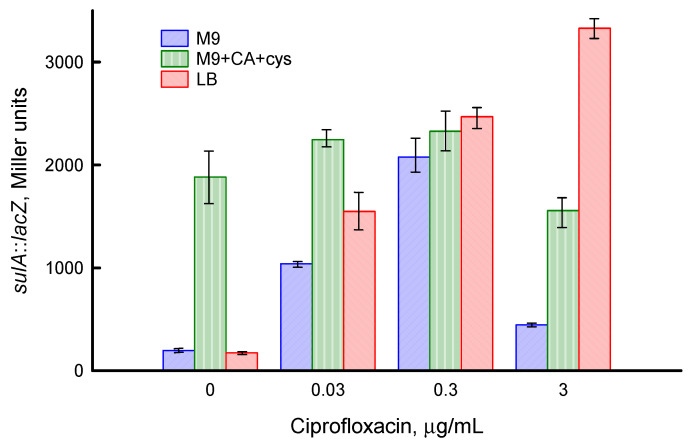
Expression of *sulA*::*lacZ* 60 min after ciprofloxacin treatment of *E. coli* growing in different media. Values are means and standard error (vertical bars) from at least three independent experiments. Statistical differences compared to control for each medium (*p <* 0.01) are noted with asterisks.

**Figure 8 biotech-12-00043-f008:**
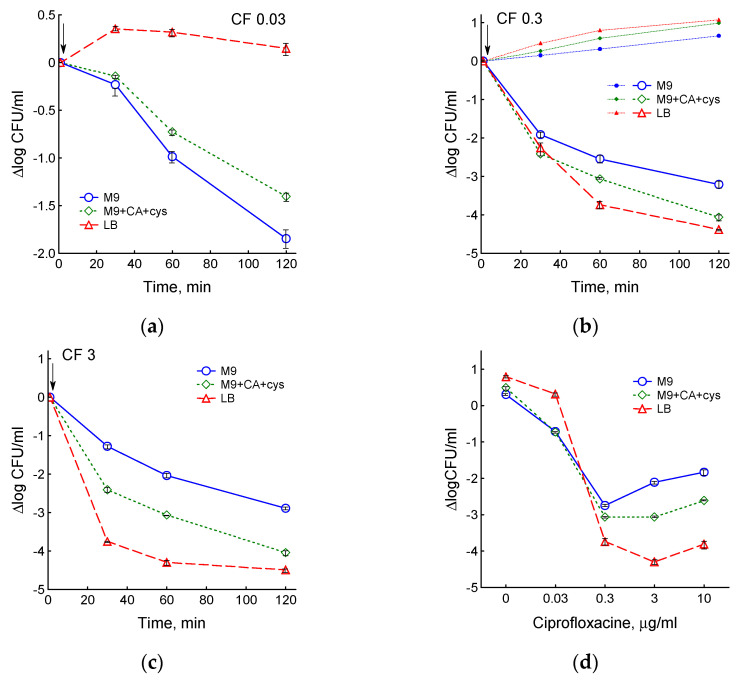
Effect of medium composition on the survival of *E. coli* treated with ciprofloxacin: (**a**) 0.03 μg/mL CF; (**b**) 0.3 μg/mL CF, small symbols and dotted lines—control; (**c**) 3 μg/mL CF. CF was added at an OD_600_ of 0.4 at the time indicated by the arrow. (**d**) The medium composition affects the optimal bactericidal concentration of ciprofloxacin. CFU data after 60 min exposure to ciprofloxacin are given. Values are means and standard error (vertical bars) from at least three independent experiments.

## Data Availability

Data used to support the findings of this study are available from the corresponding author upon reasonable request.

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
