# Peer review of "Influence of Growth Medium Composition on Physiological Responses of Escherichia coli to the Action of Chloramphenicol and Ciprofloxacin"

_biotech, 2023, doi:10.3390/biotech12020043_

Round 1

Reviewer 1 Report

The design of this manuscript is good, the research content is substantial, and the results can support the conclusion. I only have some minor comments as follow.

There need provide the statistical analysis results the for Figure 5d and Figure 6. 

In the abstract and conclusion section, there need give the significance of yhis study.

Line  9.  Change Hydrogen sulfide to "Hydrogen sulfide (H2S)".

Line 21 and line 530. It is 3 ug/mL or 3 ug/L? Please confirmed it.

Reviewer 2 Report

This article is covering some aspects of physiological responses of Escherichia coli to the action of Chloramphenicol and Ciprofloxacin.  

The specific aims of this article are exclusively directed to comparison of both antibiotics (bactericidal Ciprofloxacin and bacteriostatic Chloramphenicol with completely different mechanism of action) via physiological response of bacteria on its effects in two different media (LB) and M9) on the changes in growth, respiration, membrane potential, H2S production and bacterial survival.  Interestingly, according to the authors both antibiotics induced transient production of H2S in M9 medium. In contrast Chloramphenicol caused a sharp inhibition of metabolism in both (LB and M9) media. However, Ciprofloxacin response is strongly depended on its dose.

All the concluded observations constituted the important goals and novelty of this paper. The article is concluded with a collection of 46 mostly recent references. Additionally, all 7 Figures are very informative and with concise important data comparison and references. 

            The following suggested changes and recommendations should be introduced before the publication of the manuscript.

1.     Page 1. Abstract Line 10 & 11.  Insert abbreviation for “rich media” and “minimal media”

2.     Page 2. Line 3 from the bottom. Insert the composition of LB medium in similar way as for M9 minimal medium. line 6 from the bottom.

3.     Page 4, Section 2.5  “Study of cell viability” Insert the reference related to inhibition of galactosidase inhibition and its genotoxicity effect! 

4.     Page 6, line 7 from the bottom. Insert the figure S1 from supplementary material. This will be visually more important and convincing here.

5.     Page 8, line 4, from the bottom. Replace “stimulated” with “induced”

6.     Page 11, line 15 from the bottom. “Is two-phase”  insert “process” after “two-phase”

7.     Page 14, section 5 Conclusions.  Line 7, insert abbreviation of.. after “different media”

 The manuscript is of good quality and importance and is sequentially written and edited in order to meet the standard for the articles published in BioTech. Thus, I certainly recommend it for publication after the correction of these suggested minor changes and recommendations. 

Reviewer 3 Report

The manuscript is interesting, but the way in which the results are presented must be restructured since it could become confusing for the readers and also without a point of comparison since the behavior of the controls is presented in isolation from the treatments. Likewise, the results section should be restructured since it is a mix between results and discussion.

Some more specific points are listed below:

Abstract

L.9. Include the chemical formula of Hydrogen sulfide as it is presented in the following lines.

L.20. Include the meaning of LB and M9, please.

Introduction

L.79-79. Include the meaning of LB and M9 since it is the first time they are mentioned.

Materials and methods

L.93. M9 medium without sulfate, but with glucose…

L.97. Why did the optical density have to be at 0.4? Why did the optical density have to be at 0.4? Is there a relationship with the growth kinetics of the bacteria?

L.144. ImageJ software.

L.155. by Miller's method.

Results

The results section looks like a unified results and discussion section. Please consider that if it is a results section, it should focus on describing what was obtained in each experiment.

L.168. Change “is” to “was”.

L.189-191. Please check the wording of the sentence as it seems to be incomplete.

Figure 1. What is cam? Please Explain. What is CF?

L.203-206. This looks like from the discussion section.

L.219-221. This looks like from the discussion section.

L.220. What is SOS?

L.225. showed.

Figure 2. where is the control line? Why was M9+CA+cys medium not included for the 0.3 mcg/mL CF concentration? Why is the concentration of 3 mcg/mL included for the LB medium? The graphs are very confusing, please rearrange the data so that the effect of the treatment in each medium is obtained.

L.243. ???? This sentence does not make sense in this part of the paragraph.

L.244. The graphs present in the supplementary material only reach a time of 140 min (less than 3 hr). Please check.

L.250. in M9 (Figure S1). It is important to reference the figure so that the reader can go to see it directly.

Figure 3. To make the graphics easier to understand, it is important that the controls are also included in order to compare between treatments and controls.

L.279. It is important to specify that the M9 medium in the graphs is the one enriched with CA and cys since it could be confused with the other.

L.284. two-folds.

L.296-298. Again, these lines would correspond to the discussion section.

L.304-310. These lines look more like discussion than results section.

Figure 5. why are the results of M9+CA+cystine not presented in figure 5d?

L.339-342. This looks like the discussion of the results.

Figure 6. Why the results for chloramphenicol were not presented?

Discussion

L.405. showed.

It is clear that the composition of the culture medium has an influence on the physiological responses of E. coli when exposed to chloramphenicol or ciprofloxacin, but what is the problem that this finding solves in practice?

/

Round 2

Reviewer 3 Report

The manuscript improved significantly after the authors considered the suggestions made